

# Multi-disciplinary characterizations of the Bedretto Lab - a unique underground geoscience research facility

Xiaodong Ma[1], Marian Hertrich[1], Florian Amann[2], Kai Bröker[1], Nima Gholizadeh Doonechaly[1], Valentin Gischig[3], Rebecca Hochreutener[1], Philipp Kästli[1], Hannes Krietsch[2], Michèle Marti[1], Barbara Nägeli[1], Morteza Nejati[1], Anne Obermann[1], Katrin Plenkers[1], Antonio P. Rinaldi[1], Alexis Shakas[1], Linus Villiger[1], Quinn Wenning[1], Alba Zappone[1], Falko Bethmann[4], Raymi Castilla[4], Francisco Seberto[4], Peter Meier[4], Thomas Driesner[1], Simon Löw[1], Hansruedi Maurer[1], Martin O. Saar[1], Stefan Wiemer[1], Domenico Giardini[1]

[1]Department of Earth Sciences, ETH Zürich, Zürich, 8092, Switzerland
[2]Engineering Geology and Hydrogeology, RWTH Aachen, Aachen, 52062, Germany
[3]CSD Ingenieure AG, Liebefeld, 3097, Switzerland
[4]Geo-Energie Suisse AG, Zürich, 8004, Switzerland

*Correspondence to*: Xiaodong Ma (xiaodong.ma@erdw.ethz.ch)



## Abstract

The increased interest in subsurface development (e.g., unconventional hydrocarbon, engineered geothermal systems (EGS), waste disposal) and the associated (triggered or induced) seismicity calls for a better understanding of the hydro-seismo-mechanical coupling in fractured rock masses. Being able to bridge the knowledge gap between laboratory and reservoir

scales, controllable meso-scale in situ experiments are deemed indispensable. In an effort to access and instrument rock masses of hectometer size, the Bedretto Underground Laboratory for Geosciences and Geoenergies ('Bedretto Lab') was established in 2018 in the existing Bedretto Tunnel (Ticino, Switzerland), with an average overburden of 1000 m. In this paper, we introduce the Bedretto Lab, its general setting and current status. Combined geological, geomechanical and geophysical methods were employed in a hectometer-scale rock mass explored by several boreholes to characterize the in

situ conditions and internal structures of the rock volume. The rock volume features three distinct units, with the middle fault zone sandwiched by two relatively intact units. The middle fault zone unit appears to be a representative feature of the site, as similar structures repeat every several hundreds of meters along the tunnel. The lithological variations across the characterization boreholes manifest the complexity and heterogeneity of the rock volume, and are accompanied by compartmentalized hydrostructures and significant stress rotations. With this complexity, the characterized rock volume is

considered characteristic of the heterogeneity that is typically encountered in subsurface exploration and development. The Bedretto Lab can adequately serve as a test-bed that allows for in-depth study of the hydro-seismo-mechanical response of fractured crystalline rock masses.



## 1 Introduction

The coupled hydro-seismo-mechanical characteristics of crystalline basement rock masses have traditionally been of broad scientific and engineering interest. Fluid migration and circulation therein concerns rock mass permeability and transport, fault instability and seismicity, and ultimately crustal strength and deformability (Achtziger-Zupančič et al., 2017; Clauser, 1992; Ingebritsen & Manning, 2010; Manga et al., 2012; Townend & Zoback, 2000; Zoback & Townend, 2001). For

subsurface engineering development, fluid flow and the associated seismo-mechanical response need to be controllable (NRC, 1996). For example, in the context of engineered geothermal systems (EGS) (Tester et al., 2006), the enhancement of fluid flow typically results from fracture reactivation and seismicity. Conversely, the latter needs to be minimized concerning certain underground facilities (e.g., $CO_2$ storage, nuclear waste disposal, tunnels), in which fluid flow should be regulated or even prevented (Zoback & Gorelick, 2012).


There exists a plethora of literature dedicated to the hydro-seismo-mechanical processes taking place in single fractures (Goodman, 1989; Jaeger et al., 2007). Its fundamental mechanism has been understood as the interplay between stress, permeability and seismicity. Primarily, shear and normal stress acting on the fracture and the fracture's frictional property dictate its stability and seismicity, which consequently affect its hydraulic aperture and permeability. Crystalline basement

rock masses can often be conveniently considered as fractured systems of low-porosity, lower-permeability matrix intersected by fractures of various scales with respect to permeability and connectivity. However, it remains challenging to understand the hydro-seismo-mechanical processes in fractured rock masses (Amann et al., 2018; and references therein), because the variability and complexity therein prevent simple upscaling from single fractures.

The need to better understand the hydro-seismo-mechanical coupling in fractured rock masses becomes even more crucial in the recent context of unconventional oil and gas and deep engineered geothermal systems (EGS) and the associated (triggered or induced) seismicity (Cornet, 2015; Ellsworth, 2013; Elsworth et al., 2016; Giardini, 2009). At full-size reservoir scales, studies on the hydro-seismo-mechanical processes have to be inferred from observations at a sparse spatial resolution (e.g., Basel, Cooper Basin, Cornwall, Fenton Hill, Helsinki, Pohang, Soultz). The experiments at such scales are often

constrained by insufficient resolution in order to yield fundamental understanding and wide application. Laboratory-scale experiments, although instrumental in revealing the fundamental mechanisms, are hardly representative of the heterogeneity and complexity of natural systems such as fractured rock masses. Numerical simulations, which can model the processes at various scales, offer a great opportunity to conceptually understand hydro-seismo-mechanical processes but need to be calibrated against high resolution field observations.


The knowledge gap between laboratory and reservoir scales can be bridged through controllable meso-scale in situ experiments (Amann et al., 2018). A handful of underground research infrastructures have been either adapted from existing





mines and tunnels or newly excavated (e.g., Äspö, Canadian URL, Grimsel, Jinping, Kamaishi, KURT, Mont Terri, Reiche
Zeche, SURF) (Ingraham, 2021; Ma, 2021). The exposure of the subsurface environment offers direct access to the rock

masses at depth. This allows for sophisticated, multi-disciplinary characterization, instrumentation and experimentation at
higher spatial resolutions and more controllable scales, which otherwise would not be materialized from the surface or
through downhole instruments. Depending on site-specific conditions, various scales (from decameter to hectometer) of rock
volume can be made available for different experimental purposes, offering tailored heterogeneity and complexity.

While underground laboratories offer unique opportunities to access the rock masses in situ, unwanted effects are incurred.
The excavation inevitably perturbs the surroundings, altering the pristine rock masses and physical conditions (i.e., stress
changes in the near-field, pore pressure depletion, temperature perturbations)(Perras & Diederichs, 2016; Siren et al., 2015;
Tsang et al., 2005). Thus, the boundary conditions have to be understood and incorporated in the analysis. Acknowledging
such challenges, in situ experiments in underground laboratories remain indispensable. The efforts to approach

representative in situ conditions are still limited by the available rock mass scale, complexity and the burial depth.

A handful of in situ field experiments have been conducted in recent years (Fu et al., 2021; Hertrich et al., 2021; Ingraham,
2021; Krietsch et al., 2020; Ma, 2021; Schoenball et al., 2020), which have significantly advanced our understanding of the
hydro-seismo-mechanical processes at decameter scales; however, to what extent such experiments are representative of the

realistic in situ heterogeneous rock mass remains an open question. As an effort to step up the scale towards hectometer rock
masses (Gischig et al., 2020), the Bedretto Underground Laboratory for Geosciences and Geoenergies ('Bedretto Lab'
hereafter) was established by ETH Zürich in 2018. The existing Bedretto Tunnel (Ticino, Switzerland) has been transformed
into a state-of-the-art underground research facility, the Bedretto Lab. Various scales of experiments will be hosted here,
which are pertinent to the complex geoscience and engineering issues outlined earlier. In this paper, we formally introduce

the Bedretto Lab, its general setting and current status. The results of a first suite of multi-disciplinary characterizations are
outlined, focusing on identifying a representative rock volume. Combining the characterization efforts to date, we evaluate
the suitability of the Bedretto Lab rock volume as a test-bed to host upcoming experiments, and offer an outlook on the
challenges and opportunities to advance the understanding of hydro-seismo-mechanical processes taking place in fractured
crystalline rock masses.

## 95    2 Bedretto Lab Description

The Bedretto Lab is located in the Bedretto Tunnel in the Swiss Central Alps, near the Gotthard pass region (Figure 1a). The
Bedretto Tunnel is 5218 m long and connects the Furka Base Tunnel in the northwest with the Bedretto Valley in the
southeast (Keller & Schneider, 1982). The tunnel axis runs approximately N43°W, with a gentle slope of ~0.5% dipping
towards its south portal. The Bedretto Tunnel was excavated as part of the construction logistics of the Furka Base Tunnel to





transport the muck. The elevation at Bedretto Tunnel's south portal (Tunnel Meter, TM0) and its junction with the Furka

Base Tunnel is 1479.5 m and 1505.2 m a.s.l., respectively. Along the tunnel alignment, the rock overburden gradually rises

to a maximum of ~1632 m (corresponding to an elevation of 3124 m a.s.l.) at ~TM3140, then slightly decreases to ~1300 m

further northwest. At the location of the current Bedretto Lab (TM2000-2100), the overburden is approximately 1000 m.

The horseshoe-shaped Bedretto Tunnel was excavated by drill-and-blast with a cross-section of approximately 3 m by 3 m

to host rails for mucking. In some sections, the tunnel was enlarged to allow mucking trains to pass-by. Between TM2000-

2100, the tunnel widens into a 6 m by 3 m (width by height) niche, which was selected to host the main part of the Bedretto

Lab and the first suite of multi-disciplinary rock mass characterizations.

Since its completion in 1982, the Bedretto Tunnel remained largely unlined and unpaved, and was primarily used to facilitate

ventilation and drainage of the Furka Base Tunnel. Therefore, the rock mass structural and hydrological conditions can be

directly characterized, and the rock mass is accessible through relatively short boreholes. Detailed investigations carried out

previously focused on groundwater flow systems (Lützenkirchen, 2002; Ofterdinger, 2001), brittle fault zone structures

(Lützenkirchen, 2002), localized ductile deformation and geochronology (Rast, 2020), excavation-related rock mass failure

(Alcaíno Olivares, 2017; Ganye et al., 2020; Huber, 2004; Meier, 2017), and landslide structures (Vlasek, 2018). Since 2018,

the Bedretto Tunnel has been made available by its owner, Matterhorn Gotthard Bahn (MGB), to ETH Zürich for long-term

research, which prompted the establishment of the Bedretto Lab.

## 2.1 Geologic and tectonic setting

From its south portal, the Bedretto Tunnel consecutively penetrates metamorphic terrains of the Helvetic domain, and in

particular the Tremola series until TM434, the Prato series until TM1138, and the Rotondo granite until reaching its

northwest terminus at the Furka Base Tunnel (Keller & Schneider, 1982) (Figure 1c). The Tremola series is part of the Sasso

zone, which is characterized by the predominance of chlorite-mica schists and gneisses (Steiger, 1962). The Prato series is

characterized by amphibolites and layered biotite/quartz-feldspar gneisses in the southeast and mica/biotite gneisses and

migmatites from TM635 towards the northwest (Rast, 2020). The granite body that hosts the majority of the Bedretto Tunnel

is referred to as the Rotondo granite. The bulk composition of the Rotondo granite is primarily quartz (25-35% by volume),

alkali feldspar (microcline) (20-40%), plagioclase (albite and oligoclase) (10-25%) and biotite (3-8%) (Hafner, 1958;

Labhart, 2005). At some locations, trace amounts of mica, chlorite and garnet are encountered. The Rotondo granite is one of

several magmatic bodies of the Gotthard massif (Rotondo, Gamsboden, Fibbia, Cristallina and Medelser). The intrusion of

the Rotondo granite took place around $294 \pm 1.1$ Ma (Sergeev et al., 1995) in the late stages of the Variscan orogeny, and is

slightly younger than the Fibbia granite ($299.4 \pm 1.2$ Ma) that intruded the Gotthard massif to the northeast (Keller et al.,

1987; Schaltegger & Corfu, 1992). Ductile deformation with gneissic foliation within the Rotondo granite is generally

concentrated around a few shear zones (Schneider, 1985). In some sections, a weak foliation is encountered (Lützenkirchen





& Loew, 2011). The foliation is less pervasive than in the Fibbia granite (Schneider, 1985), where ductile shear zones are interpreted to have developed as a result of progressive Alpine deformation (Marquer, 1990). However, other studies suggest that the foliation in the older Fibbia granite developed during a late short-lived Variscan deformation phase that did not affect the younger Rotondo granite (Mercolli et al., 1994; Steiger & Guerrot, 1991).

The regional stress field near the Swiss Alps is not uniform and mainly affected by Alpine orogeny. The maximum horizontal stress ($S_{Hmax}$) azimuth is generally within the NW quadrant (Heidbach et al., 2018; Kastrup et al., 2004). Based on focal mechanism solutions of more than 100 earthquakes within the region, Kastrup et al. resolved a variation of the stress regime from a slight predominance of strike-slip in the Alpine foreland to a strong predominance of normal faulting in the high-altitude parts of the Alps. Based on the regional $S_{Hmax}$ orientation pattern, a ~NW-SE trending of $S_{Hmax}$ is expected

around the Bedretto area, which would be sub-parallel to the Bedretto Tunnel. The predominance of a strike-slip stress regime and the transition towards normal faulting in high-altitude parts of the Alps implies that reverse faulting is unlikely in the study area.

## 2.2 Structural mapping

Medium to large-scale fracture and fault zones (thickness ranging between sub-meters to tens of meters) are frequently

visible on the tunnel walls. Fabrics and mineral assemblages of brittle-ductile fault zones in the northern section of the Bedretto Tunnel (between TM3500 and TM5218) have been previously mapped and analyzed in detail (Lützenkirchen, 2002; Lützenkirchen & Loew, 2011). A complementary structural mapping has recently been conducted between ~TM1140 (near the Rotondo granite contact) and TM2800 (Jordan, 2019). Overall, fractures and fault zones within the Rotondo granite are mostly dipping steeper than 50° and an absence of structures dipping to the east and south has been noted. Figure 2 inset

shows that NE-SW (tunnel-perpendicular) and N-S striking structures dominate on stereonets. In addition, E-W and NW-SE (tunnel-parallel) sets are mapped. The tunnel-perpendicular and E-W striking sets are typically more prominent and associated with a higher degree of shearing, evidenced from core and outcrop observations. Structural mapping orientation results from the tunnel are consistent with those from surface scanline mapping in outcrops directly above the tunnel and on aerial orthophotos (Jordan, 2019).


The water inflow has also been semi-quantitatively assessed for the fractures and fault zones between ~TM1140 and TM2800 (Jordan, 2019). The water inflows into the Bedretto Tunnel are primarily associated with fractures (i.e., no visible matrix porous media flow) and fault zones. A few highly conductive fault zones are responsible for the majority of the bulk water inflow in the tunnel. In general, the tunnel-perpendicular and E-W striking sets are associated with higher inflows.

These structures often contain fault cores with gouge and cataclasites. As we will outline later, these two sets of structures can potentially be active or be activated in a strike-slip/normal-faulting stress environment, with $S_{Hmax}$ trending between E-W and SE-NW.



## 2.3 Preliminary in situ stress characterization

Along the Bedretto Tunnel, stress-induced rock failures (e.g., spalling and kinking) frequently occur on the sidewalls,
primarily in tunnel sections where pre-existing fractures are hardly present. These spalling fractures do not appear to be directly induced by tunnel excavation damage due to blasting. The appearance of these stress induced failures at the sidewalls suggests that the horizontal stress component perpendicular to the tunnel is smaller than the vertical stress. Therefore, local reverse faulting stress regime (i.e., the minor principal stress $S_3 = S_v$) is unlikely. Accordingly, $S_{Hmax}$ corresponds to either the intermediate principal stress $S_2$ (i.e., normal faulting, $S_1 = S_v$), or the major principal stress $S_1$ (i.e.,
strike slip, $S_2 = S_v$). Nonetheless, rotation of the stress tensor is possible due to local topography (Liu & Zoback, 1992; Meier, 2017). Given strong variations of the overburden above the Bedretto Tunnel (Figure 1c), the topographic effect is in competition with the tectonics-controlled stress pattern to result in significant local stress variations along the tunnel. A previous study showed that the topographic effect is strong at low overburden but diminishes significantly under larger overburden (Meier, 2017).


Small scale hydraulic fracturing tests, or mini-fracs (Haimson & Cornet, 2003), were conducted between December 2018 and July 2019 to obtain an estimation of the in situ stress field of the rock volume between TM1750 and TM2250 (Ma, Gholizadeh Doonechaly, et al., 2020). The tests were performed in six short (30 - 40 m long) SB boreholes ('SB' denoting stress measurement borehole), avoiding major fault zones (Figure 2). On borehole televiewer logs, the observed hydraulic
fractures are steeply-dipping, which generally agrees with the assumption that the overburden stress is larger than the horizontal stresses and approximates a principal stress direction. The inferred average direction of the maximum horizontal stress ($S_{Hmax}$) is approximately N100-110°E. The magnitude of the overburden or vertical stress ($S_V$) is estimated by integrating the granite density of the overburden ($\approx$ 1030 m), which is approximately 26.5 MPa. The measured $S_{hmin}$ magnitude is 14.6 ± 1.4 MPa, and the estimated $S_{Hmax}$ is 24.6 ± 2.6 MPa (Bröker, 2019; Bröker & Ma, 2021).
Acknowledging measurement uncertainty and local stress heterogeneity, the mini-frac tests indicate that the stress state in the vicinity of the Bedretto Lab is transitional between normal and strike-slip faulting conditions ($S_V \geq S_{Hmax} > S_{hmin}$). This is generally consistent with the expected regional stress state (Heidbach et al., 2018; Kastrup et al., 2004), although the stress ratio at the Bedretto Lab differs significantly from that inverted from deep earthquakes in the region.

From overnight pressure decay tests (lasting 13 - 15 hours) in the SB borehole mini-frac intervals, the pore pressure ($P_p$) was measured and range between 2.0-5.6 MPa, considerably below the expected hydrostatic pressure. This reflects the impact of tunnel drainage and pressure drawdown that has been ongoing since the tunnel's excavation. Similar underpressured conditions have also been observed at distances of 60 m (4 MPa) and 90 m (5 MPa) from previous research boreholes located near the Bedretto Tunnel's NW terminus (Evans, K., personal communication). Although such effects diminish
further away from the tunnel, studies of similar underground laboratory settings suggest that cooling and drainage associated



stress perturbations can still be present beyond 100 m from the tunnel wall (Fu et al., 2018). (Note: the ambient temperature inside the Bedretto Tunnel is ~18°C year-round.) The coupled effects of excavation damage zone, cooling and drainage certainly warrant cautious interpretation of the near-tunnel stress measurements(Evans et al., 2003).

## 3 Rock volume characterizations

In late 2019, three boreholes (CB1,2,3; 'CB' denotes characterization borehole) have been drilled between TM2000-2100 to enable a comprehensive characterization of the Bedretto Lab rock mass volume. The lengths of sub-parallel boreholes range from ~200 to 300 m, and penetrate the rock mass at the tunnel's southwest side wall. Figures 2 and 3 present the three-dimensional view of the layout of these boreholes, in relation to the tunnel and the short SB boreholes. CB1,2,3 boreholes were fully cored (with nominal borehole diameter of 97 mm and core size of approx. 63 mm). Technical details of these

boreholes are compiled in Table 1. The cores facilitated a geological interpretation of the rock volume (Section 3.1). A suite of geophysical logging runs were conducted soon after the boreholes were drilled. Figure 4 presents a composite log of CB1 as an example. Geomechanical interpretation along the penetrated rock volume is complemented by discrete hydraulic fracturing stress measurements at selected depth intervals in CB1 (Section 3.2). Geophysical imaging was made available through ground-penetrating radar (GPR) in CB1,2,3 boreholes to illuminate the complex geological structures (Section 3.3).

A multi-packer system was installed in CB2 for hydraulic characterization within the complex structures (Section 3.4). We also present some of the laboratory testing results conducted on samples of the host Rotondo granite to date (Section 3.5).

### 3.1 Geological characterization

The geologic characterization of the CB rock volume relies on the combination of core logging and acoustic/optical televiewer logs (ATV/OTV) to identify key structures. The characterized rock volume is composed of weakly deformed

Rotondo granite protolith (i.e., weekly foliated), intersected by less frequently distributed, highly foliated ductile shear zones. The mylonitic ductile shear zones are quartz- and biotite-rich and their contact with the protolith can be abrupt or gradual. The boreholes intersect a variety of structures, such as open fractures, filled fractures, the aforementioned mylonitic ductile shear zones, dikes and veins, and compositional foliation within the granite. The compilation of core description, structure typology and fault zone identification are shown in Figure 5.

Open fractures are clearly visible as traces in both ATV and OTV. In the cores, open fractures are identified by mineral precipitation on the fracture surfaces and in several instances vuggy porosity develops, likely due to hydrothermal alteration. The filled fractures are generally dark in color, commonly filled with biotite and/or quartz, and are discrete features less than 1-2 mm thick. Ductile shear zones occur in varying degrees of intensity. Some mylonite to ultra-mylonites have sub-millimeter foliations spanning a couple of meters in thickness or can occur as an abrupt strain localization feature of a couple

to tens of centimeters in thickness. The quartz (commonly smoky grey) or aplitic dikes and veins thickness ranges from less





than 1 cm to about 20 cm. While compositional foliation in the granite protolith is generally not visible in cores or logs, in some cases the grains do align to form a weakly foliated texture at the core scale.

It appears that three distinct lithological units are present in the characterized rock volume (Figure 5). Depending on the specific borehole, the first unit reaches to the measured depth (MD) of about 60 to 120 m. This unit is characterized by dikes
(mostly aplitic) and isolated shear fractures. This shallow unit of the rock volume appears rather intact with less fractured and less deformed features. Deformation is significantly more intense in the middle unit, between 120 and 200 m MD, where the majority of fault zones are located. These fault zones are composed of multiple branches of anastomosing individual fault cores. After about 100 m of this highly fractured and highly deformed unit, or below ~200 m MD, deformation seems to diminish as fault zones are fewer and thinner than in the middle unit above. The fractures in this third unit are more discrete
and singular as opposed to forming in swarms.

Fault zones were identified in the recovered cores as the combination of several features that indicate a considerable concentration of deformation. Figure 6 shows an example from the middle fault zone unit intersected by CB1. Crackle breccias and mylonites, following the fault rock classification by Woodcock & Mort (2008), are commonly identified. Fine-grained brittle fault rocks (e.g., fault gouges and cataclasites) are most probably present in the rock volume, but their poor
consolidation precludes an efficient core recovery. Our observation shows fault zones composed of multiple fault core branches with overlapping damage zones and internal lenses of rock with little deformation. This configuration is close to the conceptual model of fault zones proposed by Faulkner et al. (2003) and differs from the single fault core model of Chester et al. (1993).

Figure 7 shows the orientations of each structure type. The most important structures in terms of cumulated deformation
(lower row) trend almost exclusively NE-SW. The same pattern is shown when comparing structure orientations around fault zones with structures in between fault zones (Figure 5). Near the fault zones, the distribution of orientations tends to be unimodal around the NE-SW direction (tunnel-perpendicular) while the rock volume in between fault zones also includes structures oriented N-S and NW-SE (tunnel-parallel).

### 3.2 Geomechanical characterization

A few dedicated stress measurements were conducted at selected depth intervals in borehole CB1 via mini-frac tests. Mini-frac tests could not be conducted within the borehole measured depth (MD) of 150-250 m, as high fracture density and borehole enlargement/washouts prevented reliable packer seating and intact interval selection. The instantaneous shut-in pressures (ISIPs) of the mini-frac cycles lead to reliable estimates of the least principal stress (or presumably $S_{hmin}$) (Figure 8a). These $S_{hmin}$ values are mostly around the frictional limit imposed by a frictional coefficient of $\mu = 0.6$ and a hydrostatic
pore pressure gradient. A frictional coefficient $\mu = 0.6$ is considered to be representative for granites at depth (Byerlee, 1978). Within a few borehole intervals (e.g., MD = 53, 113, 133 m), the $S_{hmin}$ magnitudes are noticeably higher than the rest.



It is worth noting that the measured in situ pore pressure is substantially below hydrostatic; therefore, the theoretically permitted lower bound of $S_{hmin}$ in this instance should be much below the expected values under the hydrostatic conditions.

The $S_{hmin}$ values estimated from mini-fracs in CB1 are generally consistent with those obtained from the SB boreholes (Bröker, 2019; Bröker & Ma, 2021). The average $S_{hmin}$ values of SB borehole measurements are also depicted in Figure 8a, mostly near the lower bounds of the measured values in CB1 borehole. The actual stress and pore pressure gradients based on the CB1 and SB boreholes measurements facilitate the estimation of the slip tendency ($T_s = \tau/\sigma_n$) and dilation tendency ($T_d = (S_1 - \sigma_n)/(S_1 - S_3)$) (Morris et al., 1996), where $\tau$ and $\sigma_n$ are the shear stress and normal stress on the fracture/fault surface.

The slip tendency $T_s$ values associated with the CB1 fractures do not exceed 0.4, which is generally not considered critical under the typical crustal stress state.

A notable stress indicator is the occurrence of breakouts in all three CB boreholes (van Limborgh, 2020). These breakouts primarily developed within the broadly defined middle fault zone unit. Only a few breakouts were observed outside this unit.

The depths and widths of the breakouts in CB1 are summarized in Figure 8d,e. Looking downhole, the diametrically-opposite breakout pairs are systematically located around both sides of the borehole, suggesting relative strength isotropy despite weak foliations. Within the middle fault zone unit, breakouts vary in width, depth and azimuth or disappear in some sections. The breakouts' azimuthal rotations immediately near individual fractures are likely associated with stress perturbation due to shear dislocation of fossil or active fractures/faults (Shamir & Zoback, 1992); the longer wavelength

rotations spanning the entire fault zone unit plausibly reflect systematic stress variation associated with the fault zone. The breakouts rotate counter-clockwise (looking downhole with the top of the borehole referenced as north) by ~50° between ~145 m and 175 m MD, i.e., the beginning to the middle of the fault zone unit, then rotate backwards by approximately the same extent until they terminate at the end of the fault zone unit (at ~220 m MD).

### 3.3 Geophysical imaging

Geophysical imaging of the Bedretto Lab rock volume consists of ground-penetrating radar (GPR) in both single-hole and cross-hole configurations. The sensitivity of electromagnetic waves is affected by different rock properties, namely the dielectric impedances between the host rock and faults/fractures. GPR surveys have been conducted in all three CB boreholes with antenna systems of various center frequencies (20, 100, 250, 500 and 1000 MHz) and varying spacings.

The premise of GPR reflection imaging is to image structures that provide a contrast in dielectric properties in the medium.

In the Bedretto Lab rock volume, this is primarily a contrast between fractures (filled by clayey minerals and/or water) and the granitic host rock. Laboratory measurements on borehole cores reveal that the Rotondo granite has little to no variability in dielectric properties, and is largely isotropic. The loss-tangent (phase angle between the resistive and reactive components) of the dielectric constant is small, which facilitates large penetration depths. The relative dielectric permittivity





of the host rock ($\varepsilon_r$ = 5.5) does not vary over the applied frequency range. Single-hole reflection imaging, where both

transmitter and receiver antennas are in the same borehole, provides clean and repetitive data that can be used to extract geometric information about the major fault zones present in the characterized rock volume. For the detailed analysis procedure, we refer to Shakas et al. (2020).

The electrically resistive granitic rock of the Bedretto Lab is a pristine setting for GPR reflection imaging. Figure 9 shows the 100 MHz reflection survey for borehole CB1,2,3. Clear reflections arising from nearby boreholes, as well as from several

potentially water-filled fractures and faults were identified. The first major fault intersects the borehole CB1 at approximately 145 m MD, which is consistent with the ATV/OTV logging observations. This fault provides a strong reflector that is traced over several hundreds of meters. The observed thickness of the reflected structures on the GPR image correlates with their areas and hydraulic apertures.

The chevron type (V-shaped) pattern that the reflector (Figure 9) exhibits is a known ambiguity of borehole GPR surveys.

This artifact is introduced by projecting the fault/fracture plane that intersects the borehole in 3D onto 2D space(Olsson et al., 1985). To overcome this issue, Hediger (2020) performed the correlation between the structures inferred from GPR reflections and ATV/OTV data, in an effort to delineate the major fault zones and fractures. The consistency between both data sets further constrained the geometry of those features, minimizing the potential bias introduced by individual datasets (Molron et al., 2020).


### 3.4 Hydraulic characterization

Hydraulic tests were carried out in borehole CB1,2,3 to characterize their transmissivity and connectivity (Münger, 2020). Borehole CB1 and CB3 were closed at the borehole mouth. In CB2, a multi-packer system was used to effectively isolate six intervals (see Fig 3b). The CB2 intervals were chosen based on the observed fracture/fault clusters from the core and logging

observations (Table 2): individual fractures in intervals 1 and 2, frequent occurrence of fractures in intervals 3 and 4, and fault zones/fracture zones in intervals 5&6 and 7. For hydraulic characterization, constant flow rate tests were carried out in CB1 and CB3 as well as in all six intervals in CB2. Repeated tests were performed in intervals 4 and 7 in CB2. All test results, including the repeated measurements, are summarized in Table 2.

The estimated transmissivities for different intervals/boreholes differ by several orders of magnitude. CB1 and CB3, each characterized along its full length, show the highest transmissivities (~$2.1 \times 10^{-6}$ m$^2$/s and ~$4.5 \times 10^{-7}$ m$^2$/s, respectively). However, the transmissivities of individual boreholes have to be treated with caution, since these long intervals include several conductive structures with non-uniform pressure head, a condition not consistent with the analytical analysis method applied. The isolated intervals in CB2 can be classified into three different groups based on their increasing transmissivities:





a) intervals 1 and 2, b) intervals 3 and 4, and c) intervals 5&6 and 7. The estimated transmissivities in CB2 intervals are consistent with the geological observations. Since all three boreholes are sub-parallel, CB1 and CB3 are expected to encompass the majority of the fractures/faults included in intervals 1 to 7 in CB2. Therefore, it is expected that the transmissivity values of CB1 and CB3 are at least as high as the largest transmissivity observed within the intervals of CB2, which is indeed the case.


In order to identify major hydraulic flow pathways within the characterized rock volume, individual constant flow rate tests (drawdown/buildup) were conducted in CB1 and CB3 boreholes. The pressure response was monitored in the other borehole and in all CB2 intervals. The pressure response time is defined as the first notable pattern change in the pressure signal in the monitoring intervals and boreholes since the drawdown/buildup. The drawdown tests were executed with a constant

extraction flow rate of 120 Liter/hour in CB1, and 90 Liter/hour in CB3. Each drawdown test was followed by a buildup test (Note: the characterization radius during the buildup test can be limited by the accuracy of the pressure gauge and the duration of prior drawdown interval and flow rate (Bourdarot, 1999)). Based on the pressure decline curves, the characteristic response time between different boreholes/intervals during the drawdown tests are estimated (Table 3).

As shown in Figures 10, during the drawdown test in CB1, all six CB2 intervals and the CB3 were hydraulically connected to CB1. However, during the drawdown test in CB3, the pressure response was only observed in CB2 intervals 3, 4, 5&6 and 7 and in CB1. The results also show significant heterogeneity within the test volume. For example, interval 7 in CB2 shows strong hydraulic connectivity to CB3, with a response time of approximately 7 min, which contrasts the pressure response time of about 1 hour during CB1 drawdown. Interval 5&6, which is located immediately below interval 7 in CB2, shows a

very rapid hydraulic response to CB1 drawdown (less than 2 min), but a significantly delayed response to CB3 drawdown (~50 min). Intervals 1 and 2 in CB2 are hydraulically connected with CB1, with a response time of approximately 100 min, however no hydraulic response was observed after ~180 min of drawdown in CB3. Given these observations, none of the intervals in CB2 seems to exhibit comparable hydraulic connectivity with CB1 and CB3, and a systematic pattern was not identified.

**3.5 Laboratory petrophysical and mechanical characterization**

Based on visual inspection, the majority of the Rotondo granite exposed at the tunnel wall appears to be homogeneous and isotropic. In the deeper parts of the CB boreholes, ductile shearing is apparent, suggesting physical anisotropy. Current laboratory benchtop characterizations (on various petrophysical and mechanical properties) were mostly focusing on the visually homogeneous core samples. The results suggest a low to moderate elastic anisotropy combined with considerable

non-linearity of the elastic response. Table 4 gives a list of the physical and mechanical properties of the Rotondo granite in dry and water-saturated conditions. The details for these measurements are documented in (David et al., 2020).





Despite its isotropic appearance and the absence of apparent fabric orientation, ultrasonic-wave velocity measurements indicate that the Rotondo granite is moderately anisotropic, with the *P*-wave anisotropy factors of about 6% and 20% for dry and water-saturated samples, respectively. Considerable surge in the ultrasonic-wave velocity by saturation (more than 50%), significant non-linearity in the stress-strain relationship, high permeability and considerably low *P*-wave quality-factor of 4.9 (i.e., high attenuation level), all suggest a highly micro-cracked structure of the Rotondo granite.

The Rotondo granite features higher permeability when unconfined, as compared to other types of known granites. The permeability of Rotondo granite in the characterized Bedretto rock mass is roughly 10 times higher than that of Grimsel granite, and 100 times larger than that of Westerly granite (Brace et al., 1968; David et al., 2020; Wenning et al., 2018). The *P*-wave velocity is considerably dependent on the confinement pressure, suggesting a highly micro-cracked structure (David et al., 2020). If the high micro-crack density is characteristic of the pristine Rotondo granite in situ, significant poroelastic response is expected given elevated pore pressures.

## 4 Interdisciplinary interpretations of the rock volume

The multi-disciplinary characterization of the Bedretto Lab conducted so far identified a rock volume that is both scientifically interesting and practically representative. The fractures and fault zones intersected by the CB boreholes inform us of the strong structural complexity and spatial heterogeneity at multiple scales. This is evidenced by the individual observations within and between several boreholes and different borehole intervals. Below we strive to provide an interdisciplinary interpretation of the characterization results, particularly in the context of the suitability of the rock mass as a test-bed to better understand the hydro-seismo-mechanical response of realistic crystalline basement rock reservoirs.

### 4.1 Heterogeneous rock mass, representative test volume

The characterized rock mass volume encompasses a multitude of features. One of the most prominent features is the middle unit composed of major fault zones and sandwiched by two comparatively more intact units. Although this middle fault zone unit is composed of several fault branches, it is found that these branches are generally sub-parallel to each other and form a cluster (Figures 5 and 7). The whole cluster potentially traces back to the tunnel wall and coincides with the major fault zone observed between TM 1950-1993 (Castilla et al., 2020). Major fault zones of this scale seem to be repeatedly present along the Bedretto Tunnel for every few hundreds of meters (Schneider, 1985), and they are generally trending perpendicular to the tunnel (NE-SW) and/or E-W. Therefore, the presence of the middle fault zone unit, along with the sandwiching units, reasonably characterizes the rock mass that could be encountered within the Bedretto Lab.





The sandwiching units above and below the middle fault zone unit are also considered to be characteristic of the Rotondo granite protolith. The two sandwiching units seem relatively homogeneous and share similar appearance, mineralogy (inferred from spectral gamma logs) and physical properties (e.g., wave velocities). Their properties are also consistent with those of the rock volume characterized by the SB boreholes scattered along the Bedretto Tunnel (Caspari et al., 2019; Greenwood et al., 2019). For example, the velocity profile along borehole CB1 (Figure 4) shows gradual increase of $V_P$ and $V_S$ with depth (from ~5250 m/s and up, comparable to ~5400 m/s from the SB borehole logs and the laboratory core measurements), despite the anomalies associated with the intersection of the major fault zones.

What accompanies the lithological unit variations is the stress variations along the CB boreholes. Although a more complete stress profiling is yet to be conducted, the azimuthal rotation of the breakouts across the middle fault zone unit informs us of the changes in stress orientations and magnitudes. The study to quantify why the breakouts only develop within the major fault zone but not in other parts of the CB boreholes nor any SB boreholes is currently ongoing. Plausibly, low rock strength in the fault zone (substantially lower than the intact rock core) can promote breakout development. According to our scoping analysis, the breakout azimuth at ~145 m and 220 m MD in CB1 corresponds to a far-field $S_{Hmax}$ azimuth between E-W and SE-NW (Zhang & Ma, 2021), which is generally consistent with the average value of ~N110°E measured from several SB borehole mini-fracs. The breakout rotation towards the middle of the major fault zone reaches ~50°, which requires substantial stress rotation and reduction in relative stress difference (or stress ratio $\phi$). This could only be accommodated by the gradual changes in fault zone lithology and the associated rheological variations (Casey, 1980; Faulkner et al., 2003, 2010).

The stress orientation reversal towards the end of the middle fault zone unit indicates that it is likely to revert to the expected far-field stress condition that has been characterized. The local and global rotations of the breakouts suggest various scales of stress perturbations, which warrant further modeling. The stress variations simply manifest the heterogeneity and complexity of the rock volume. Such convoluted lithological and stress heterogeneity are characteristic of realistic fractured rock masses, and should be considered when designing and conducting hydro-seismo-mechanical experiments therein.

**4.2 Prevailing structures, hydraulically-conductive features**

The major structure sets in the Bedretto Lab rock mass are all present in the characterized rock volume. There are four prevailing sets of fractures/faults identified along the Bedretto Tunnel (azimuth N43°W). All four sets of structures have been intersected by characterization boreholes CB1,2,3 (azimuth N133°W). We are cognizant of potential undersampling of certain structures in each mapping campaign. For example, the tunnel-parallel sets might be under-mapped along the tunnel, and similarly for the NE-SW striking sets along the CB boreholes. However, this does not seem to be the case for the CB boreholes (inset of Figure 2), as there are abundant structures striking ~NE-SW (±15°), sub-parallel to or at acute angles with





the borehole azimuth. This is attributed to the ~45° inclination of these boreholes so that the undersampling of these steeply-
dipping structures is remedied to some extent. It suffices to state that the prevailing sets of fractures/faults in the Bedretto
Lab are reasonably represented in the characterized rock mass volume.

As alluded to earlier, the NE-SW and E-W striking sets of fractures and fault zones appear to be the primary structures that
are hydraulically-conductive in the Bedretto Lab rock volume. These structures have been identified from the tunnel walls,
contributing to relatively higher inflow rates among other sets. This qualitative correlation is confirmed by several
independent lines of evidence noted in the CB borehole characterization. Coinciding with these fracture/fault sets,
appreciable anomalies have been identified along the thermal and electrical conductivity logging profiles (Figure 4b,c); the
core samples exhibit significantly higher degree of shearing; strong reflections are shown on GPR images, indicating
relatively wider hydraulic apertures and/or higher dielectric property. These observations all suggest that the NE-SW and E-
W striking sets are the main hydraulically-conductive conduits in the Bedretto Lab rock volume.

It is worth noting that the NE-SW and E-W striking sets are more favorably-oriented in the prevailing normal and/or strike-
slip faulting regime. Taking the measured average of N100°E $S_{Hmax}$ azimuth, steeply-inclined structures forming acute angles
with respect to $S_{Hmax}$ are generally more susceptible to slip. Quantitatively, the calculated slip tendency shown in Figure 8
indicates that the NE-SW and E-W striking sets are indeed associated with higher slip tendency. Although the absolute
values of slip tendency (< 0.4) are below the empirical frictional limits (~0.6) (Byerlee, 1978), the relative criticality between
different structure sets seems to support the first-order control of the in situ stress.

It has been generally regarded that critically-stressed fractures and faults are associated with hydraulic conductivity (Barton
et al., 1995; Townend & Zoback, 2000), because the naturally-occurring hydro-shearing processes enhance and maintain
fracture permeability. The critically-stressed fracture concept can plausibly explain the NE-SW and E-W striking sets being
more hydraulically-conductive, applicable to both the tunnel-mapping and CB borehole structures. Previous field
observations supporting the critically-stressed fracture concept (Barton et al., 1995; Rogers, 2003) were mainly conducted at
scales of several kilometers long full-size boreholes so that this first-order relationship is not heavily affected by local stress
variabilities that occur at shorter wavelengths. While this might be the case for the tunnel-mapping structures, it is perhaps
tenuous to justify in the case of the CB structures. As already shown, strong stress variations are evident along the CB
boreholes, particularly around the fault zones. The local stress variations inevitably affect the slip tendency of individual
fault branches. Given that the fault-perturbed in situ stress state becomes less anisotropic, the slip tendency is expected to
decrease, weakening the critically-stressed fracture concept. The associated stress changes around the fault zone further
complicate the correlation between the stress criticality and fracture conductivity for individual fractures/faults.
Nevertheless, it is important to take into account the corresponding scale when the stress variability is concerned (Ma, Saar,



et al., 2020). The applicability of the critically-stressed fracture concept to the particular case here certainly warrants further study.

Alternatively, stress-controlled hydraulic-conductivity can be evaluated based on the dilation tendency (Morris et al., 1996). This concept was introduced for crustal rock masses at relatively shallow depths (e.g., <1 km) (Mattila & Follin, 2019), for which variations of the normal stress on the fracture/fault exert significant control on its hydraulic aperture, and consequently, conductivity. The calculated dilation tendency profile along borehole CB1 (Figure 8c) shows that the main conductive structures are subject to high normal stress, i.e., low dilation tendency, which seems to weaken the dilation

concept. It is ambiguous to quantify the dilation tendency of tunnel-mapping structures, as the exact stress condition is unknown and subject to significant topographic variations.

Correlating stress with hydraulic conductivity assumes that the present stress state dominates. However, the high-conductivity feature of certain structure sets might have already developed under the paleo-stress condition. Although the

stress condition has evolved, the high conductivity could still sustain until present days. If that is the case, distinguishing them from those structures naturally reactivated and hydraulically-enhanced in geologically recent time would be challenging.

## 4.3 Complex, compartmentalized hydro-structures

Along the Bedretto Tunnel, recurring major fault zones serve as the main hydraulic conduits, channelizing fluid circulation in the rock mass. Since these fault zones are generally sub-parallel, it is unknown to what extent these main conduits are hydraulically connected. Preliminary hydrological and geochemical analysis indicates that water composition changes between these conduits (Brixel, B., personal communication), which suggests certain degrees of hydraulic compartmentalization of the whole rock mass along and across major structures. Such hydraulic compartmentalization also

exists within the rock volume characterized by the CB boreholes. During the drilling phase, it was reported that abrupt increases of formation pore pressure and flow rate were associated with the penetration of the middle fault zone unit and branches therein (Meier, 2020).

Apparently, those fault zones in the Bedretto Lab rock volume simultaneously act as the main hydraulic conduits along the

fault planes and as impermeable layers across the fault planes. This is consistent with the general understanding of the fault structure in that the fault core is surrounded by damage zones (Chester et al., 1993; Faulkner et al., 2003). The fault core can be relatively impermeable for cross-flow but is able to maintain overpressure and appreciable flow therein (Faulkner et al., 2010). There was significant core loss and borehole enlargement when those fault zones were penetrated, so only a



qualitative understanding of the fault structure was possible from examining cores and borehole televiewer logs (Figures 5
and 6). On the other hand, GPR profiles allowed us to infer the physical contrast between the protolith and the fault zone
rocks (Figure 9). Strong reflections of the fault zones due to distinct water-bearing capacity clearly set themselves apart from
the Rotondo granite, although recognizing the exact fault trace is challenging, which is due to the inherent ambiguity of the
GPR interpretation and the complexity of the intersecting fault (zone) branches.

The complexity of the major fault zones results in compartmentalized hydro-structures. Hydraulic characterization in the CB
rock volume revealed significant heterogeneity of hydraulic transmissivity (Table 2). Such heterogeneity is present both
along individual boreholes and between boreholes, depicting a complicated hydraulic backbone of the rock volume. The
hydraulic transmissivities differ by several orders of magnitude along multiple packed intervals of borehole CB2. This
reflects the significant discrepancy of hydraulic property between several permeable fractures/fault zones segmented by the
multi-packer system. An interesting observation is the asymmetric hydraulic response between both sides of CB2, i.e., a
diametrically-opposite behavior between the CB1-CB2 and CB3-CB2 connectivity. As suggested earlier, correlation of cores
between the CB boreholes suggests that the major fault zone varies in thickness and features multiple laterally-inconsistent
branches (Figures 5 and 6). This could explain the irregularity of spatial hydraulic compartmentalization and asymmetric
hydraulic response within the rock volume. The local irregularity of structure geometry and the stress perturbation associated
with the fault zones may also exert additional influence. A better understanding of the hydro-structures and the hydro-
mechanical response within the rock volume requires carefully planned tracer tests and geophysical imaging, which is
beyond the scope of this paper.

## 5 Concluding remarks

The Bedretto Lab has recently been established in the Swiss Central Alps on the basis of the existing Bedretto Tunnel. It
serves as an underground geoscience research laboratory and geoengineering test-bed. The Bedretto Lab represents the state-
of-the-art for conducting meso-scale experiments on the crystalline rock masses and offers opportunities for international
collaborations. The Bedretto Lab is now fully operational and its main granitic rock mass volume has been extensively
characterized via multi-disciplinary approaches. Combined geological, geomechanical, hydrogeological and geophysical
methods were employed in several hectometer-scale boreholes to probe the in situ conditions and internal structures of the
rock volume. A scientifically interesting and practically representative rock volume has been identified.

The characterized rock volume is approximately 100 m by 300 m by 100 m in size, off the southwest sidewall of the
Bedretto Tunnel between TM2000-2100. The rock overburden there exceeds 1000 m, and the stress environment is
dominated by normal and/or strike-slip faulting. The rock volume features three distinct units, with the middle fault zone
sandwiched by two relatively intact units. The major fault zone appears to be a representative feature of the site, as similar





structures repeat every several hundreds of meters along the Bedretto Tunnel. The fault zones are visible both on extracted cores and borehole imaging tools. The lithological variations across the fault zone manifests the complexity and heterogeneity of the rock volume. Significant variations of the hydrological and mechanical properties at various scales are evident. Pronounced stress rotations across the fault zone are observed. Compartmentalized hydrostructures have been

identified, which seem to be segmented by the major fault zone and branches therein.

The characterized rock volume encompasses a multitude of complex features, and it approximates the representative scale and heterogeneity typically encountered in subsurface exploration and development of basement rocks. The rock volume will be further characterized and densely instrumented with tailored sensors. It will allow for in-depth studies of the hydro-

seismo-mechanical response of fractured rock masses. The characterized rock volume will host a series of customized hydraulic stimulation experiments, serving as a test-bed for EGS reservoirs (referred to as the Bedretto Reservoir Project, BRP). Another rock volume further down the Bedretto Tunnel will be subsequently characterized and made available, enabling sophisticated fault reactivation experiments to study induced seismicity (referred to as the Bedretto Earthquake Project, BEP). These upcoming experiments are full of challenges and opportunities, with the hope to bridge the current

knowledge gap and offer new insights.



**Author contribution**

All authors of this paper collectively contribute as a team of the Bedretto Underground Laboratory for Geosciences and Geoenergy. The role of each team member is described here on the Bedretto Lab website. Please see through the following
link. http://www.bedrettolab.ethz.ch/about/team/

**Competing interests**

The authors declare that they have no conflict of interest.

**Acknowledgements**

The Bedretto Underground Laboratory for Geosciences and Geoenergy is an ETH infrastructure and is financed by ETH
Immobilien. The Bedretto Lab experiments are funded by the Swiss Federal Office of Energy (SFOE) (project VALTER), by the EU Horizon 2020 (project DESTRESS), by the EU initiative Geothermica – EraNet (project ZoDrEx and project SPINE), the Werner von Siemens Stiftung (project MISS) and by ERC (project FEAR). The Bedretto tunnel is property of the Matterhorn Gotthard Bahnen (MGB). Help from Simone Zaugg and Shihuai Zhang on figure editing is greatly appreciated.




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



**Tables**


Table 1. List of characterization boreholes and measurements conducted therein.

| Borehole # | Location (TM) | Diameter (mm) | Length (m) | Inclination (degree) | Logging performed | Additional tests |
|---|---|---|---|---|---|---|
| CB1 | 2050 | 97 | 303 | 45 | ATV, OTV, GPR, CAL, Cond., DEV, FWS, SGAM, Temp. | Mini-frac |
| CB2 | 2043 | 97 | 220 | 40 | ATV, OTV, GPR, CAL, Cond., SGAM, Temp. | Pressure monitoring |
| CB3 | 2037 | 97 | 192 | 50 | ATV, OTV, GPR, CAL, Cond., SGAM, Temp. | |

Note:

1. All borehole azimuths are oriented N133°W. The nominal borehole diameter is based on the 97 mm coring bit; the actual borehole diameters slightly exceed 97 mm, and vary with the coring scheme.

2. ATV/OTV: Acoustic/Optical televiewer; GPR: Ground-penetrating radar; CAL: Caliper; Cond.: Electrical conductivity; DEV: deviation tool; FWS: Full-waveform sonic; SGAM: Spectral Gamma; Temp.: Temperature.






Table 2. Single-hole Hydraulic Test Results

| Interval / Borehole # | Interval Depth (MD) (m) | Interval / Borehole Length (m) | Test Date (in 2020) (mm.dd) | Transmissivity ($m^2$/s) | | | | Initial Pressure (MPa) |
| | | | | *(Theis, 1935)* | | *GRF (Barker, 1988)* | | |
| | | | | *Drawdown* | *Buildup* | *Drawdown* | *Buildup* | |
|---|---|---|---|---|---|---|---|---|
| CB1 | - | 303 | 03.14 | $1.5 \cdot 10^{-6}$ | $1.4 \cdot 10^{-6}$ | $2.8 \cdot 10^{-6}$ | $2.2 \cdot 10^{-6}$ | 4.02 |
| CB3 | - | 192 | 03.13 | $4.1 \cdot 10^{-7}$ | $4.0 \cdot 10^{-7}$ | $5.7 \cdot 10^{-7}$ | $3.7 \cdot 10^{-7}$ | 4.05 |
| CB2 | intervals | | | | | | | |
| 1 | 199.8-221.8 | 22.2 | 03.07 | $1.4 \cdot 10^{-10}$ | $1.4 \cdot 10^{-10}$ | $8.7 \cdot 10^{-11}$ | $8.3 \cdot 10^{-11}$ | 4.04 |
| 2 | 196.8-198.3 | 2.0 | 03.12 | $4.1 \cdot 10^{-11}$ | $4.4 \cdot 10^{-11}$ | $1.4 \cdot 10^{-11}$ | $1.2 \cdot 10^{-11}$ | 4.06 |
| 3 | 177.2-195.2 | 18.5 | 03.05 | $1.1 \cdot 10^{-8}$ | $8.4 \cdot 10^{-9}$ | $8.6 \cdot 10^{-9}$ | $6.9 \cdot 10^{-9}$ | 3.90 |
| 4 | 167.7-175.7 | 9.5 | 03.09 | $1.4 \cdot 10^{-8}$ | $1.5 \cdot 10^{-8}$ | $9.5 \cdot 10^{-9}$ | $4.9 \cdot 10^{-9}$ | 3.94 |
| 4 | - | 9.5 | 03.11 | $1.7 \cdot 10^{-8}$ | $1.3 \cdot 10^{-8}$ | $2.0 \cdot 10^{-9}$ | $5.4 \cdot 10^{-9}$ | 3.97 |
| 4 | - | 9.5 | 03.11 | $1.2 \cdot 10^{-8}$ | $1.2 \cdot 10^{-8}$ | $4.8 \cdot 10^{-9}$ | $3.4 \cdot 10^{-9}$ | 3.98 |
| 5&6 | 141.7-165.2 | 24.1 | 03.11 | $8.4 \cdot 10^{-8}$ | $8.5 \cdot 10^{-8}$ | $1.3 \cdot 10^{-7}$ | $4.1 \cdot 10^{-8}$ | 3.99 |
| 7 | 125.1-140.1 | 15.5 | 03.03 | $2.1 \cdot 10^{-7}$ | $1.8 \cdot 10^{-7}$ | $5.4 \cdot 10^{-8}$ | $4.6 \cdot 10^{-8}$ | 3.62 |
| 7 | - | 15.5 | 03.06 | $1.2 \cdot 10^{-7}$ | $2.0 \cdot 10^{-7}$ | $2.7 \cdot 10^{-8}$ | $4.2 \cdot 10^{-8}$ | 3.67 |

Note:
1. The packer between intervals 5 and 6 did not provide proper sealing, resulting in a direct hydraulic connection between
the two intervals. Thus, the interconnected intervals 5 and 6 are considered a single interval, i.e., 'interval 5&6'.
2. Pressure measurements were conducted at the tunnel floor. Thus, the hydrostatic heads at the (center of the) interval depth
are subtracted from the reported pressure values.
3. The analysis of the transient pressure curves was carried out with the MATLAB Toolbox 'hytool' (Renard, 2017). The
tests were analyzed with two models, Theis (1935) and Generalized Radial Flow (GRF) (Barker, 1988).
4. The initial pressures of the boreholes/intervals were also determined with Horner (1951) plots and linear fitting.



Table 3. Characteristic pressure response time in the monitored boreholes/intervals during the drawdowns

| Interval/Borehole # | Response time (hh:mm:ss) during the drawdown in | |
|---|---|---|
| | CB1 | CB3 |
| CB1 | - | 00:44:54 |
| CB3 | 00:56:42 | - |
| CB2 intervals | | |
| 1 | 01:41:42 | - |
| 2 | 01:14:48 | - |
| 3 | 00:26:47 | 02:08:24 |
| 4 | 00:07:02 | 00:50:54 |
| 5&6 | 00:01:55 | 00:51:54 |
| 7 | 05:16:42 | 00:06:38 |






Table 4. Selected physical properties of the Rotondo granite (measured under no confining stress)

| Property | Specification (unit) | Dry | Water-saturated |
|---|---|---|---|
| Porosity | connected (%) | 1.36 | - |
| | total (%) | 1.75 | - |
| Density | grain (kg/m$^3$) | 2653 | - |
| | bulk (kg/m$^3$) | 2606 | 2620 |
| Ultrasonic velocity | $P$-wave, $V_P$ (km/s) | 3510 | 5434 |
| | $S$-wave, $V_S$ (km/s) | 1785 | 2526 |
| Elastic modulus (dynamic) | Young's modulus (GPa) | 22.6 | 45.9 |
| | bulk modulus (GPa) | 27.3 | 60.5 |
| | shear modulus (GPa) | 8.3 | 16.8 |
| | Poisson's ratio | 0.36 | 0.37 |
| Permeability | ($\mu$D) | - | 4.35 |
| Tensile strength | Brazilian Test (MPa) | 8 | - |
| Compressive strength | Uniaxial (MPa) | 172 | - |
| Fracture toughness | Mode I (tensile) (MPa·m$^{1/2}$) | 1.3 | - |
| | Mode II (shear) (MPa·m$^{1/2}$) | 4 | - |








**Figure 1: a) Integrated geological, topographical and seismological information near the Aar and Gotthard massif surrounding the Bedretto Lab (Keller and Schneider, 1982; adapted from Lützenkirchen and Loew, 2011, and Gischig et al., 2020). b) Map view of the Bedretto Tunnel and the hosting Rotondo granite. c) Cross-sectional view of the Bedretto Lab with respect to the tunnel.**




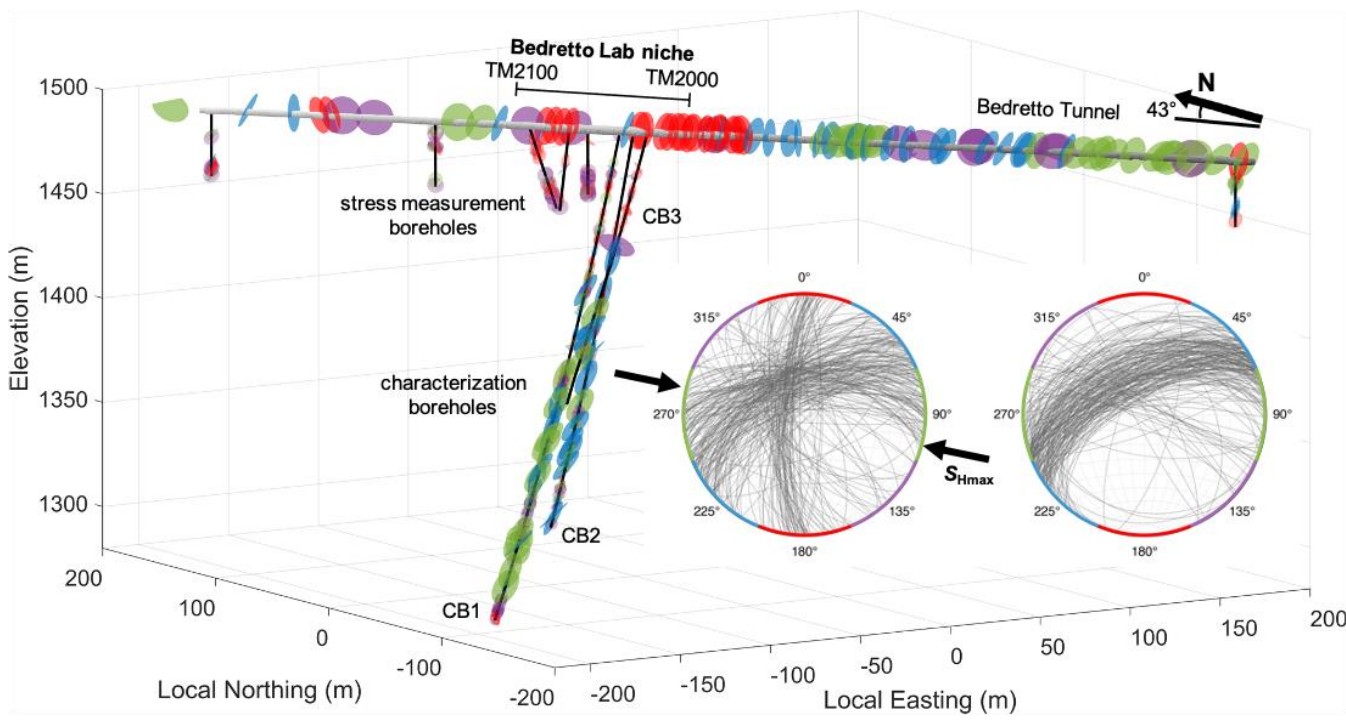

**Figure 2: Configuration of the CB1,2,3 boreholes with respect to the Bedretto Tunnel. Fractures and fault zones, mapped along the tunnel and the boreholes, are colored with respect to their strikes. Stress measurement boreholes (SB) are also shown. Inset: Stereonet of fractures and fault zones mapped along the tunnel (left) and the CB1,2,3 boreholes (right) (only structures represented in the lower row of Figure 6), respectively. The four fracture/fault sets are colored distinctly according to their strikes, which is also marked on the circumference of the stereonets.**



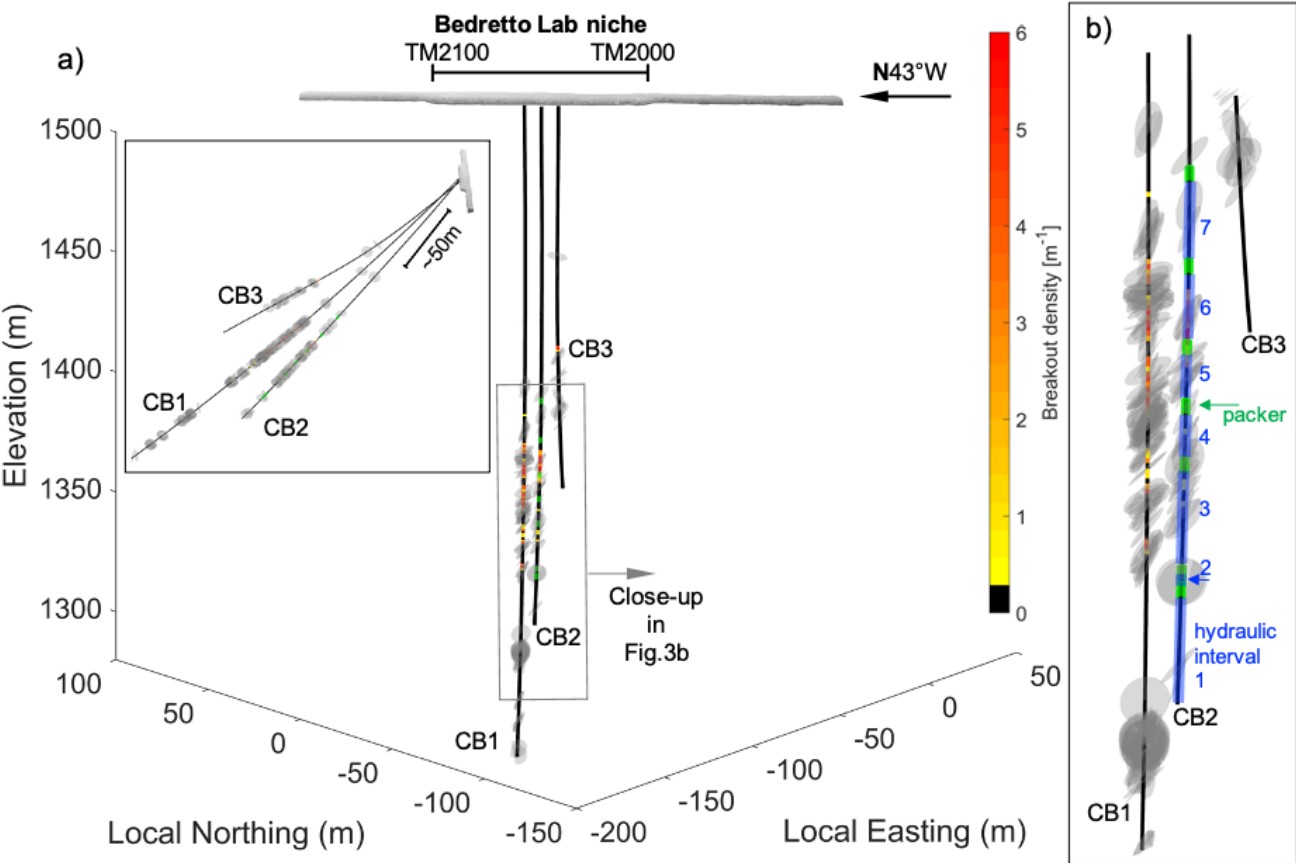

**Figure 3: Configuration of boreholes CB1,2,3 and the mapped major fractures/fault zones therein. a) Looking down approximately normal to the three boreholes. Inset shows the side view of the boreholes. b) Close-up of the major fault zone interval. Note the breakout density along the CB borehole major fault zone and the multi-packer system (and the divided hydraulic intervals) installed in CB2 (see Table 2 for details).**






**Figure 4: Integrated geophysical logs of borehole CB1: gamma ray; electrical conductivity; temperature; sonic compressional and shear wave velocities ($V_p$ and $V_s$); density of mapped fractures from televiewer logs.**






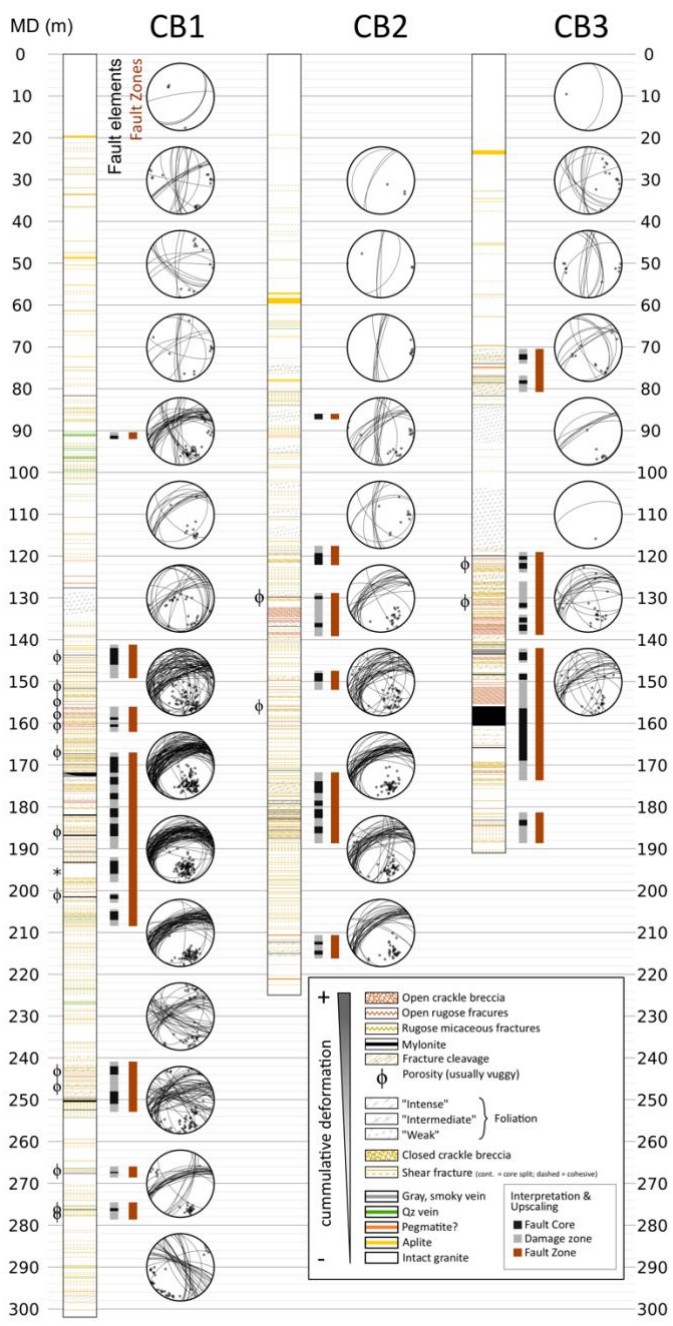

**Figure 5: Core description showing the geological designations and structural orientations (measured by the acoustic and optical televiewers, i.e., ATV/OTV).**




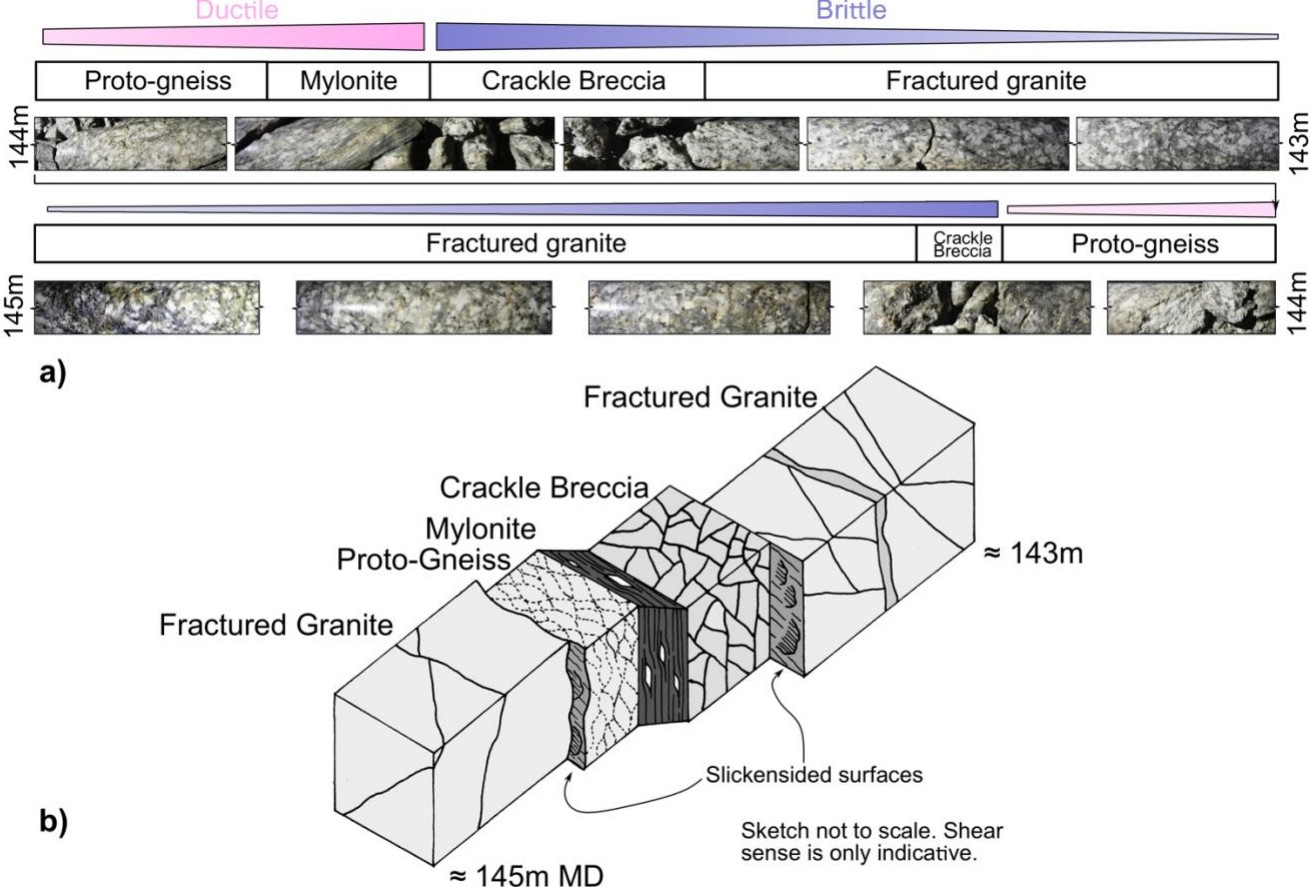

**Figure 6: a) Core images, compositions and the structure of the major fault zone encountered in borehole CB1 (between 143-145 m measured depth). Ductile and brittle structures can be seen in close proximity to each other. Brittle structures are located at the boundaries of ductile features; b) Schematics of the fault zone structure.**






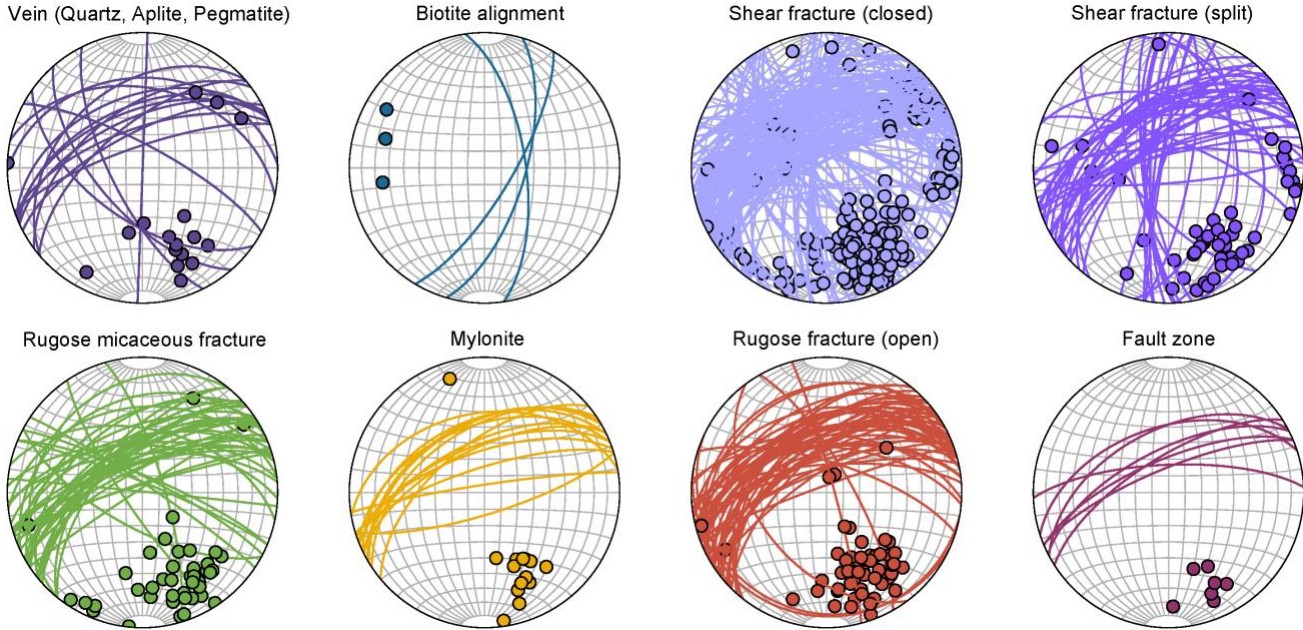

**Figure 7: Mapped geological structures by type. The upper row shows the orientation of structures that are mainly 'closed' (based on visual examination of the cores). The lower row shows the orientations of structures with higher shear strain and can be perceived as 'open'. Note the structures shown in the lower row are almost exclusively oriented NE-SW.**





**Figure 8: Integrated geomechanical information of borehole CB1: stress and pore pressure profiles (Diamonds are from SB borehole data; circles are from CB1, while open circles are of uncertainty; dashed blue gradient represents hydrostatic pore pressure.); slip tendency ($T_s$) and dilation tendency ($T_d$) of all mapped structures (shown in Figure 5); breakout azimuths and widths (0° refers to the high-side of the borehole).**


**Figure 9: Single-hole GPR reflection profiles (100 MHz) along: a) CB1, b) CB2, and c) CB3. From each borehole, the nearby boreholes are identified (and labelled) as prominent reflectors. The middle unit of major fault zones are clearly visible.**






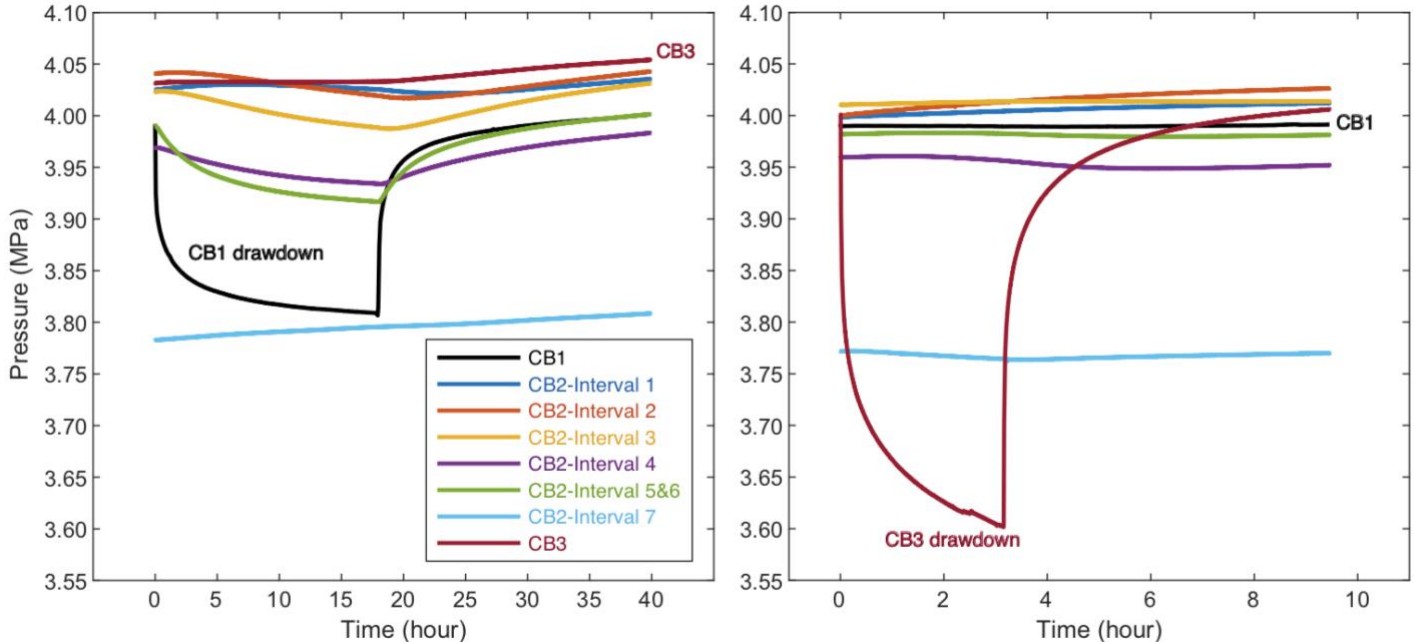

**Figure 10: The pressure response of boreholes/intervals to the drawdown in CB1 (left) and CB3 (right).**
