# Peer review of "Multi-disciplinary characterizations of the Bedretto Lab"

_Solid Earth, 2021_

## Referee Comment (RC3)

Review comments from PhD Robert Earon, specialist in hydrogeology at SKB, Sweden:

- While I understand that this type of project involves a lot of collaboration and many parties and hence the length of the author list, it would be beneficial to briefly explain in the text the contributions of the various groups.
- Line 47-48: Surely if there is a plethora of research there can be more up to date references?
- Line 68: it is not certain that all readers will be familiar with all facilities. Countries or regions should be included in this list.
- Line 74: I'm unclear on what is meant here regarding tailored heterogeneity and complexity.
- Line 156: Please elaborate on what is meant by semi-quantitatively.
- Line 158-159: this is a common effect of the Cubic Law. It would be beneficial to quantify this claim or provide a reference.
- Line 190-191: These pressure decay tests need a bit more explanation or a reference.
- Line 191: Please provide the in-situ hydrostatic pressure at the lab elevation (or groundwater level for reference)
- Line 245 and Figure 2: this is probably one of the largest issues with the study, in that the 3 longest boreholes are all oriented in roughly the same direction. It is unsurprising that the fracture data is weighted to fractures and fracture sets which the boreholes would be geometrically more likely to intercept. Any claim regarding knowledge of the heterogeneity of fracture-related geoscientific data must be carefully examined in this light. Fracture data should be treated as inherently biased based according to the orientation of the boreholes, even when corrected.
- Line 295: Is it possible that the change in the dielectric permittivity seen in the GPR survey could be due to mineral fillings such as graphite?
- Line 301-304: I think that readers would appreciate a more thorough interpretation of the GPR profiles. What are all the reflectors above the fault zone? Do they correspond with air reflections (i.e. through checking the geometry of the parabolas using) or do the parabolic shape and depth correspond with velocities in rock? What are the actual distances to the fault zones and do they correspond with your conceptual understanding of the site?
- Line 307-313: What are the length of the packers? How do you characterize connectivity? What are the durations of the tests and magnitude of the flow rates? I find the description of the hydrogeological testing methods needs considerable elaboration.
- Line 317-319: What was the analytical method for analysis and why is this violated by the pressure gradient?
- Line 330: What was the corresponding head change and was pseudo-steady state achieved?
- Line 335-344: It is unsurprising that the fracture network is heterogeneous and anisotropic. However, the times of responses are difficult to interpret without the estimated distance between the sections. It seems from Figure 9 that the boreholes intercept a major fault. Where are the sections with regards to the fault? The Structural geological information is absolutely vital in interpreting the results of the interference tests. Did you match the drawdowns against analytical solutions using i.e. Aqtesolv? How do the hydraulic properties compare with the ones of the packer tests? This could be vital in separating the near-field effects (closed fractures or fracture clusters which give high transmissivities during transient hydraulic tests) from the actual connected hydraulic properties.
- Figure 1: Please include a contour map. The topography is vital in understanding the placement and orientation of the tunnels and boreholes.
- Figure 4: Are all fractures treated similarly? It would be interesting to see the open vs. closed fracture count and perhaps plot major fault zones which were intercepted.

- Figure 10: Please normalise this figure so that simply drawdown is shown. (Assuming either a water density or simply showing the change in pressure. CB2 interval 1 looks odd, and usually one isn't concerned with the absolute pressure in these types of tests.
- Line 410: I'm relieved that the authors have included this, but this this should have been mentioned far earlier. Additionally, why was no effort made to correct for the bias? The authors mention that several fractures were found at acute angles to the borehole orientation, but what is the fracture intensity? These could be the fracture sets with highest true intensities.
- 414-415: the logical conclusion of this claim is that a borehole oriented orthogonally to the 3 existing boreholes would provide an entirely new data set and give a better understanding of the fracture matrix properties.
- 415-416: This claim needs to be carefully motivated. At present it lacks sufficient evidence to be included in the article.
- 454-455: I'm not a rock mechanics expert, but I believe that given a reasonably stiff rock at depth dilation due to stress is mitigated by the rock matrix itself. However, in proximity to stress gradients like tunnels and the ground surface the effects will be far more prominent. I would suggest removing the claim regarding the weakening of the dilation concept.
- Line 474: I think the word "apparently" is misused. However, I agree with the claim. Often the fracture core may have clay gouge, fillings etc which inhibit transverse flow.
- Line 487: I'm unfamiliar with the term "hydraulic backbone"
- Line 492-494: I think the authors are correct, although the more care and perhaps a figure (in 3 dimensions) showing the hydraulic diffusivity of the sections and the location of the fault would make the point clearer. I believe the structure of the major zones is the underlying cause for the compartmentalization the authors indicate, but a bit more work needs to be done to really support the argument.

---

## Author Response (AR1)

Re: Review of Solid Earth manuscript se-2021-109

Dear Dr. Healy and the Solid Earth editorial board:

Thank you for handling the review of our manuscript referenced above. We appreciate the opportunity to further improve our work. We have now carefully revised the manuscript according to all comments. Please find our updated version (in track-changes and with changes-incorporated) enclosed. Appending to this letter, we provide our response to the comments made by the reviewers, and refer to the specific changes we made throughout the text.

In the revised manuscript, all of these issues have been carefully addressed with corrections wherever they apply. We believe that these changes further help to enhance our paper.

Thank you for your assistance. Please do not hesitate to contact us should you have any questions.

Xiaodong Ma (on behalf of all co-authors)
* * *
**Review by RC1:**

We thank Referee#1 for the positive comments! Our replies are detailed below and will be reflected in the revised manuscript.

Line 43 Clarify what "latter" refers to.

Response:
'latter' here refers to 'fracture reactivation and seismicity'. This is clarified in the revised version.

Line 104. It looks like the FURKA tunnel is horseshoe-shaped, not the Bedretto Tunnel.

Response:
The shapes marked in Fig.1 might be ambiguous. We've modified it.

Line 200. Replace "have been" with were.

Response:
Addressed.

Line 300. Add a space after "space".

Response:
Addressed.

Line 333. Define "characteristic response time"

Response:
Addressed. 'Characteristic response time' here refers to the duration between the moment of the drawdown/pressurization test start and the moment of measurable pressure signal associated with the test.
* * *
**Review by Pär Grahm (RC2):**

**(By Section)**
In general,
the paper: Well written, interesting and well referenced description of the performed rock characterizations. In addition, a good summary of the Bedretto Lab establishment

Response:
We thank the reviewer for the positive comments. We reply to your specific comments item by item below.

Orientation system:
It becomes evident that the text is written by many authors when it comes to which orientation system that is used. Sometimes N ##°W are used, other tImes NW-SE striking, and sometimes stríke/dip, and even trend for fracture/zones. Different disciplines do use different systems as standard, but try to minimize the use of systems in the same paper.

Response:
We thank the reviewer for raising this issue. We revised the text to adhere to the N###°E convention for the fracture/fault strike, tunnel orientation and horizontal stress component azimuth. It might be confusing when the four fracture/fault sets are mentioned, which we arbitrarily divide into four categories (E-W, N-S, SE-NW and NE-SW).

Uncertainty:
For the stress measurements there is some uncertainty span provided, but what about all other parameters; have you done any uncertainty evaluation of magnitudes and orientations? Multiple measurements are needed to get a good picture of the values interpreted. The only text where it is stated that multiple test have been performed is at row 312-313

Response:
For the uncertainty of stress magnitudes and orientations, we refer to a dedicated stress study by (Bröker & Ma, 2021). While additional tests are still warranted to further consolidate the stress uncertainty in the vicinity of the Bedretto Lab, Line 175-190 provides our best estimation to date.

Title: A unique underground geoscience research facility
Not obvious that a reader absorbs what is the uniqueness of the facility. Few external comparisons are made with international URLs and just a brief benchmark with Grimsel regarding permeability of the granite, see also comment section 500-502. Instead of using "unique" in the title I suggest Bedretto should be presented as "a new underground geoscience research facility"

Response:
We appreciate that the reviewer points to this issue. We have toned down this and used 'new' instead in the title.

Line 68: Äspö
Äspö HRL

Response:
Addressed.

Line 137: 'of more than 100 earthquakes within the region' Time interval?

Response:
Addressed.

Line 144: 'Medium to large-scale fracture'
Consider introducing the definition Full Perimeter Intersection "FPI" as used by SKB and other repository engineering companies, e.g. SKB R-06-05 "Using observations in deposition tunnels to avoid intersections with critical fractures in deposition holes"

Response:
We thank the reviewer for sharing this reference. However, we were unable to retrieve this one online. Instead, we identified a relevant reference, which we will try to implement in our future work.
Raymond Munier, 2010. Full perimeter intersection criteria – Definitions and implementations inf SR-Site. SKB TR-10-21, Svensk Kärnbränslehantering AB.
https://inis.iaea.org/collection/NCLCollectionStore/_Public/41/095/41095194.pdf

Line 228: 'Three distinct lithological units are present in the characterized rock volume (Figure 5) '
Figure 5 is a not an interpretation of the rock volume but core descriptions from three boreholes?

Response:
Clarified.

Line 322-324: 'Therefore, it is expected that the transmissivity values of CB1 and CB3 are at least as high as the largest transmissivity observed within the intervals of CB2, which is indeed the case'
"it is expected" and "indeed the case" in the same sentence?

Response:
Revised.

Line 410-415: 'Risk of undersampling'
The configuration of the lab (orientations of tunnel + boreholes) may underestimate intensity of fractures and structure that have pole trend/plunge around 45/45 (strike/dip 135/45) It would therefore be of great interest to drill a hole with azimuth 45 and inclination -45 (from horizontal plane)

Response:
We thank the reviewer for this valuable suggestion, which is stressed by Dr. Earon (RC3, see below). We are aware of the risk of under sampling of fracture sets in the Bedretto Lab volume. Additional boreholes have been planned and will provide relevant information.

Line 440: 'wavelengths'
Maybe use higher frequency?

Response:
Revised.

Line 441-442: "As already shown, strong stress variations are evident along the CB boreholes, particularly around the fault zones."
Where is this already shown?

Response:
The stress variations we referred to concern the prominent rotations of borehole breakouts observed in all three boreholes. This is first described in Line 268-279, and also noted in Line 390-400.

Line 500-502: 'The Bedretto Lab represents the state-of-the art for conducting meso-scale experiments on the crystalline rock masses and offers opportunities for international collaborations' "State-of-the-art" methods have indeed been used for the characterization of the local rock volume. However, as a novel laboratory this is too much of an exaggeration at this stage. There are several underground research laboratories in the crystalline rock with many decades of experience and solid scientific results, e.g Äspö HRL in Sweden where site investigations began 1986.
Other aspects that appear to be missing for the Bedretto Lab to be a "state-of-the-art" test bed are an open database with access to e.g. various data from continuous hydro monitoring and water chemistry development over time, a 3D-modelled interdisciplinary site description to e.g. assist in deciding on new survey sites as well as a customized service organisation at the site.

Line 518-519: 'The rock volume will be further characterized and densely instrumented with tailored sensors. It will allow for in-depth studies of the hydro-seismo-mechanical response of fractured rock masses.' Possibly, it can be presented with even greater clarity what the Bedretto Lab can offer the future research customers, see comment 500-502.

Response:
We appreciate your comments on the 'state-of-the-art' description, which we have toned down and made relevant revisions. We are aware of many existing underground labs and the invaluable research findings generated there. These serve as great encouragement for our future work!

Table 4: 'units'
I guess that the unit for the p and s waves should be m/s and not km/s, or the numbers should be divided by 1000?

Response:
Addressed.

Figure 2: 'reference'
The caption reference to the "lower row of figure 6". I guess it should reference to figure 7.

Response:
Corrected. We thank the reviewer for spotting this!

Figure 8: 'Labelling'
The text reference e.g. Figure 8a, 8b etc, but there is no such labels in Figure 8.

Response:
Addressed.
* * *
**Review by Robert Earon (specialist in hydrogeology at SKB, Sweden) (RC3):**

- While I understand that this type of project involves a lot of collaboration and many parties and hence the length of the author list, it would be beneficial to briefly explain in the text the contributions of the various groups.

Response:
We thank the reviewer for raising this issue. According to the requirement of the journal, we will provide a statement of contributor roles (preferably adhering CRediT Taxonomy) on our lab website.

- Line 47-48: Surely if there is a plethora of research there can be more up to date references?

Response:
We have included additional references in the updated manuscript.

- Line 68: it is not certain that all readers will be familiar with all facilities. Countries or regions should be included in this list.

Response:
Addressed.

- Line 74: I'm unclear on what is meant here regarding tailored heterogeneity and complexity.

Response:
We revised with 'desired' instead.

- Line 156: Please elaborate on what is meant by semi-quantitatively.

Response:
We revised with 'qualitatively' to be precise.

- Line 158-159: this is a common effect of the Cubic Law. It would be beneficial to quantify this claim or provide a reference.

Response:
We are unsure whether the reviewer refers to 'A few highly conductive fault zones are responsible for the majority of the bulk water inflow in the tunnel.', or the previous sentence. In either case, it is based on our field observation, which is clarified in the revision. We are happy to provide further revision or explanation if the reviewer gives more specific information.

- Line 190-191: These pressure decay tests need a bit more explanation or a reference.

Response:
Addressed.

- Line 191: Please provide the in-situ hydrostatic pressure at the lab elevation (or groundwater level for reference)

Response:
There is limited information on the actual measured groundwater level in the vicinity of the Bedretto tunnel. Vlasek (2018) conducted relevant study and the estimated groundwater level directly above the Bedretto Lab (Tunnel Meter ~2000m) is 700-900m. We have revised the relevant text and provide the reference there.

(Note: We group the following four comments here to respond to them collectively.)

- Line 245 and Figure 2: this is probably one of the largest issues with the study, in that the 3 longest boreholes are all oriented in roughly the same direction. It is unsurprising that the fracture data is weighted to fractures and fracture sets which the boreholes would be geometrically more likely to intercept. Any claim regarding knowledge of the heterogeneity of fracture-related geoscientific data must be carefully examined in this light. Fracture data should be treated as inherently biased based according to the orientation of the boreholes, even when corrected.

- Line 410: I'm relieved that the authors have included this, but this should have been mentioned far earlier. Additionally, why was no effort made to correct for the bias? The authors mention that several fractures were found at acute angles to the borehole orientation, but what is the fracture intensity? These could be the fracture sets with highest true intensities.

- 414-415: the logical conclusion of this claim is that a borehole oriented orthogonally to the 3 existing boreholes would provide an entirely new data set and give a better understanding of the fracture matrix properties.

Response:
We thank the reviewer for pressing on this issue. Besides the three characterization CB boreholes, the short SB boreholes and the Bedretto Tunnel (shown in Figure 2) are orientated substantially different from the former. The SB boreholes and the tunnel, to some extent, sample fractures of certain orientations that could have been missed by CB boreholes. However, due to the limited lengths of SB boreholes, they probably are of local importance only. Additional boreholes with different orientations have been drilled since the completion of CB boreholes, which showed fracture sets and frequencies practically consistent with the CB borehole and tunnel mapping observations. More boreholes are planned to be drilled soon. We hope to address this issue with more confidence in the near future.

- 415-416: This claim needs to be carefully motivated. At present it lacks sufficient evidence to be included in the article.

Response:
We have toned down this statement and noted the necessity of future work to verify this claim.

- Line 295: Is it possible that the change in the dielectric permittivity seen in the GPR survey could be due to mineral fillings such as graphite?

Response:
Indeed, all reflections arise from contrasts in dielectric permittivity and this could be either water-filled structures or mineral fillings. We have recently submitted a separate study where we combine borehole televiewer data with the GPR reflections to assess this assumption, and found that the reflections are mainly driven by contrast in water and granite. We now refer the reader to this separate manuscript (please see comment below).

- Line 301-304: I think that readers would appreciate a more thorough interpretation of the GPR profiles. What are all the reflectors above the fault zone? Do they correspond with air reflections (i.e. through checking the geometry of the parabolas using) or do the parabolic shape and depth correspond with velocities in rock? What are the actual distances to the fault zones and do they correspond with your conceptual understanding of the site?

Response:
Thank you for pointing this out. We have now rephrased the geophysical-characterization section to address this comment and the one above. The newly added text reads as follows:
"
A more detailed study that combines GPR reflections and televiewer observations to delineate the geometry of the observed major fault, can be found in Shakas et al. (2021). By comparing televiewer observations to near-borehole GPR effects, the latter study also suggests that the observed reflections are primarily due to water-filled (open) structures (faults and fractures) and not to mineral-filled (closed) structures. We further notice that the GPR reflections match well the assumed geometry of the major fault and can further introduce constraints on the fault geometry further away from the boreholes.

The chevron type (V-shaped) pattern that the reflector (Figure 9) exhibits is a known ambiguity of borehole GPR surveys. This artifact is introduced by projecting the fault/fracture plane that intersects the borehole in 3D onto 2D space (Olsson et al., 1985). To overcome this issue, Hediger (2020) performed the correlation between the structures inferred from GPR reflections and ATV/OTV data, in an effort to delineate the major fault zones and fractures. Furthermore, several diffractions can be seen in the upper volume. These are most probably due to water-filled fractures/faults that are sub-perpendicular to the borehole trajectory (Grasmueck et al., 2010).
"

- Line 307-313: What are the length of the packers? How do you characterize connectivity? What are the durations of the tests and magnitude of the flow rates? I find the description of the hydrogeological testing methods needs considerable elaboration.

Response:
Thanks for the comments. The details are now added to the revised manuscript.
- The isolating part of the packers are one-meter long.
- The connectivity is analyzed by injection/production in one interval or borehole and checks the measurable pressure response in all others.
- The test flow rates are designed based on a preliminary estimate from a pulse test in the corresponding intervals in order to have (ideally) a maximum pressure change of 1 MPa to minimize geomechanical effects during the hydraulic testing.

- The duration of the flow or recovery test is set in such a way to observe infinitely acting radial flow.

- Line 317-319: What was the analytical method for analysis and why is this violated by the pressure gradient?

Response:
The methodology and criteria to analyze the flow test results as well as the assumptions are now described in more details within the manuscript.
- Depending on the pressure transient curve profile and the diagnostic plots, either Theis (Theis 1935) or GRF (Barker 1988) models were used to estimate the transmissivity and storativity of the tested interval/borehole. (as previously indicated in Table 2 notes)
- The isolated intervals in CB2 as well as the openholes (CB1, CB3) are assumed to be at steady state pressure before the start of the flow tests. However, this assumption, and therefore, the estimated transmissivities of individual boreholes have to be treated with caution, since these long intervals, in particular the open holes, include several conductive structures with non-uniform pressure headshead which might cause some cross flow between different structures within the same test intervals.

- Line 330: What was the corresponding head change and was pseudo-steady state achieved?

Response:
The pressure magnitudes in CB1 and CB3 changed by 0.2 and 0.4 MPa at the end of the flow period. The flow periods were long enough to observe infinitely acting flow regimes (for at least 1.5 log cycle after the wellbore storage effect subsides, as a rule of thumb). The flow test in CB1 did not show any boundary effect at the end of the flow period, whereas CB3 showed signs of an infinite linear constant head boundary at the end of the flow period. These details are now added to the revised manuscript (see Section 3.4).

- Line 335-344: It is unsurprising that the fracture network is heterogeneous and anisotropic. However, the times of responses are difficult to interpret without the estimated distance between the sections. It seems from Figure 9 that the boreholes intercept a major fault. Where are the sections with regards to the fault? The Structural geological information is absolutely vital in interpreting the results of the interference tests. Did you match the drawdowns against analytical solutions using i.e. Aqtesolv? How do the hydraulic properties compare with the ones of the packer tests? This could be vital in separating the near-field effects (closed fractures or fracture clusters which give high transmissivities during transient hydraulic tests) from the actual connected hydraulic properties.

Response:
The distance between all the packed intervals in CB2 to boreholes CB1 and CB3 are listed in the table below for a better comparison of the hydraulic response time.

Table 1. Shortest distance with connecting feature between CB2 intervals with boreholes CB1 and CB3

| CB2 | Shortest distance to CB1 (m) | shortest distance to CB3 (m) |
|---|---|---|
| Interval 1 | 23.8 | 60.4 |
| Interval 2 | 23.4 | 59.9 |
| Interval 3 | 21.4 | 50.7 |

| | | |
|---|---|---|
| Interval 4 | 23.6 | 45.4 |
| Interval 5&6 | 18.1 | 29.1 |
| Interval 7 | 17.1 | 30.3 |

The distances are estimated based on extrapolating and measuring the shortest fracture/fault between two intervals/boreholes. The shortest connection between CB1 and CB3 is also estimated to be approximately 29.7 m.

Based on the results of the GPR surveys, which are presented in Fig. 9, although the presence of a major cross-cutting structure that intersects all three boreholes (CB1,2,3) is evident from the survey, it is not fully comparable with the result from hydraulic tests. For example, the observed major structure from Fig. 9b intersects borehole CB2 at Interval 7, whereas results from hydraulic tests show strong hydraulic connection only between Intervals 5&6 in CB2 with borehole CB1, but not with CB3. This can be mainly attributed to the heterogeneities in the reservoir volume.

The analysis of all the transient pressure curves was carried out with the 'hytool' MATLAB Toolbox (Renard, 2017) with two models, Theis (1935) and Generalized Radial Flow (GRF) (Barker, 1988). The utilized toolbox uses analytical tools to fit the pressure transient profile with the selected radial flow models.

The manuscript is now revised accordingly at the corresponding sections (see Section 3.4).

- Figure 1: Please include a contour map. The topography is vital in understanding the placement and orientation of the tunnels and boreholes.

Response:
Here we refer to the work by (Meier, 2017). A simple but fundamental study regarding the effect of topography on the tunnel placement and stress state around the Bedretto Lab was documented there. This reference is already included in our paper.

- Figure 4: Are all fractures treated similarly? It would be interesting to see the open vs. closed fracture count and perhaps plot major fault zones which were intercepted.

Response:
The following/attached fracture histogram shows three different fracture categories: open fractures, filled fractures (corresponds to closed fractures, these are generally infilled by biotite and/or quartz), and undifferentiated fractures. The last category represents fractures which cannot be identified with certainty whether they are open or filled. The major fault zones are represented by an increase of the open fracture count, and numerous with varying extent are intersected by the boreholes. The consistent picking of the fault zones, their classification, and correlation between boreholes is still ongoing and involves data sets that image the fault zones away from the borehole wall, e.g. GPR measurements. We will report the results in one of our upcoming publications.

[Figure]

- Figure 10: Please normalise this figure so that simply drawdown is shown. (Assuming either a water density or simply showing the change in pressure. CB2 interval 1 looks odd, and usually one isn't concerned with the absolute pressure in these types of tests.

Response:
Please see the figure attached below. The relative pressure changes are provided in a normalized fashion. This figure is included in the revised manuscript.

[Figure]

- 454-455: I'm not a rock mechanics expert, but I believe that given a reasonably stiff rock at depth dilation due to stress is mitigated by the rock matrix itself. However, in proximity to stress gradients like tunnels and the ground surface the effects will be far more prominent. I would suggest removing the claim regarding the weakening of the dilation concept.

Response:
Agree. We have toned down the statement there.

- Line 474: I think the word "apparently" is misused. However, I agree with the claim. Often the fracture core may have clay gouge, fillings etc which inhibit transverse flow.

Response:
Agree. We have revised this expression.

- Line 487: I'm unfamiliar with the term "hydraulic backbone"

Response:
The term backbone had been mainly used within the manuscript to refer to the dominant hydraulic pathway. For better clarity, the corresponding sentence now reads as: "Such heterogeneity is present both along individual boreholes and between boreholes, depicting complicated dominant flow paths within the rock volume."

- Line 492-494: I think the authors are correct, although the more care and perhaps a figure (in 3 dimensions) showing the hydraulic diffusivity of the sections and the location of the fault would make the point clearer. I believe the structure of the major zones is the underlying cause for the compartmentalization the authors indicate, but a bit more work needs to be done to really support the argument.

Response:
We appreciate the reviewer for pressing on this issue. Beyond the phase of this characterization work, additional boreholes have been drilled to allow for a more detailed mapping and hydraulic tests of the fractures/faults in the volume. The analysis is ongoing and we hope to report the latest in the near future, which could shed some light on this issue.

**References**

Bröker, K., & Ma, X. (n.d.). Estimating the least principal stress in a granitic rock mass: systematic mini-frac tests and elaborated pressure transient analysis. *Rock Mechanics and Rock Engineering*. https://www.research-collection.ethz.ch/handle/20.500.11850/466482

Meier, M. (2017). *Geological characterisation of an underground research facility in the Bedretto tunnel* [ETH Zurich]. https://doi.org/10.3929/ethz-b-000334001

Vlasek, A. (2018). *Deep structures of large toppling slopes at the Bedretto Adit (Ticino, Switzerland)*. ETH Zürich.